# Transforming Health Care Delivery towards Value-Based Health Care in Germany: A Delphi Survey among Stakeholders

**DOI:** 10.3390/healthcare11081187

**Published:** 2023-04-20

**Authors:** Franziska Krebs, Sabrina Engel, Vera Vennedey, Adrienne Alayli, Dusan Simic, Holger Pfaff, Stephanie Stock

**Affiliations:** 1Institute of Health Economics and Clinical Epidemiology (IGKE), Faculty of Medicine and University Hospital Cologne, University of Cologne, 50935 Köln, Germany; 2Faculty of Human Sciences and Faculty of Medicine, Institute of Medical Sociology, Health Services Research, and Rehabilitation Science (IMVR), University of Cologne, 50933 Köln, Germany

**Keywords:** value-based healthcare, implementation, healthcare systems

## Abstract

Value-based healthcare (VBC) represents one strategy to meet growing challenges in healthcare systems. To date, VBC is not implemented broadly in the German healthcare system. A Delphi survey was conducted to explore stakeholders’ perspectives on the relevance and feasibility of actions and practices related to the implementation of VBC in the German healthcare system. Panellists were selected using purposive sampling. Two iterative online survey rounds were conducted which were preceded by a literature search and semi-structured interviews. After two survey rounds, a consensus was reached on 95% of the items in terms of relevance and on 89% of the items regarding feasibility. The expert panels’ responses were in favor of the presented actions and practices of VBC in 98% of items for which consensus was found (n = 101). Opposition was present regarding the relevance of health care being provided preferably in one location for each indication. Additionally, the panel considered inter-sectoral joint budgets contingent on treatment outcomes achieved as not feasible. When planning the next steps in moving towards a value-based healthcare system, policymakers should take into account this study’s results on stakeholders’ perceptions of the relative importance and feasibility of VBC components. This ensures that regulatory changes are aligned with stakeholder values, facilitating greater acceptance and more successful implementation.

## 1. Introduction

The German health care system is one of the most expensive in the world. Health care expenditure as a share of national gross domestic product is the second largest among all Organisation for Economic Co-operation and Development (OECD) countries and the largest within Europe [1]. In 2019, health care expenditure amounted to 411 billion Euros, an increase of 4.9% over the previous year [2]. New strategies and reforms of the healthcare system are needed to rein in expenditures and improve health outcomes at the same time.

Value-based care (VBC) is a concept first described by Porter and Teisberg [3]. Within the concept, the authors describe maximizing value for patients as the overriding goal of healthcare systems. Value is defined as health outcomes achieved per dollar spent, with outcomes being clinical results most relevant for patients [3]. The authors emphasize that the focus should be on improving value for patients, rather than solely lowering costs [3].

For the implementation of VBC, six core strategies were defined by Porter and Lee [4]. These comprise *(1) Measuring outcomes and costs for every patient*, *(2) Organizing into integrated practice units*, *(3) Integrating care delivery across separate facilities*, *(4) Expanding excellent services across geography*, *(5) Moving to bundled payments for full care cycles* and *(6) Implementing an enabling information technology platform* [4].

The *organization of care in integrated practice units (IPUs)* implies that a team of clinical and non-clinical health workers organizes their care around a clinical condition to provide the full care cycle for this condition. The *measurement of outcomes as well as costs for every individual patient* describes that outcomes and costs measured should span the full care cycle and outcomes measured should focus on those relevant to patients. Data on outcomes and resource use is intended to drive competition for value between providers. The implementation of *bundled payments for full care cycles* includes both outpatient and inpatient services as well as rehabilitative services provided within an IPU or care network. *Integrating care delivery across separate facilities* involves placing a higher focus on selecting an appropriate level of care with health care services organized according to complexity and the degree of specialization. *Expanding excellent services across geography* should be pursued by forming networks and facilitating professional exchange. Lastly, an *enabling information technology platform* should be implemented that digitally records all patient-relevant data and is made accessible to all providers involved in the care of a respective patient. Thereby integrated and multidisciplinary care is supported. These six strategies are presumed to be mutually reinforcing one another [4].

As illustrated above, VBC targets multiple aspects of a healthcare system and thus requires diverse actions and practices in various domains of the healthcare system to be successfully implemented. Yet, the concept, as proposed by Porter and colleagues, is framed and described as a strategic vision without supplying specific guidance on how to proceed in practice. Moreover, actions needed for the implementation of VBC depend on the specific circumstances and prerequisites in the respective healthcare system [5]. As a result, approaches to implementation vary between healthcare systems and countries with best practices regarding specific characteristics of a health system remain largely unclear [5,6,7]. 

However, actions targeted towards implementing VBC in various health systems have been shown to be most favorable if bottom-up initiatives are politically supported [7]. Hence, acceptance of and attitude towards the concept among those stakeholders that are affected by system changes and involved in legislation is critical for the implementation of VBC. Even though the implementation of VBC is already being advanced in other countries [5,8,9] or at the supranational level [10], little knowledge exists on implementation specifically in the German context. The previous literature examined the German healthcare providers’ knowledge of VBC, but only focused on the hospital sector [11]. There exists a knowledge gap regarding the perspective of various German healthcare stakeholders on the specific steps of implementing the VBC approach.

Therefore, the aim of this study was to assess stakeholders’ perceptions and to identify expert consensus on the relative relevance and perceived feasibility of actions and practices for moving towards a value-based healthcare system in Germany.

## 2. Materials and Methods

The study was conducted in 2021/2022 and consisted of a preparatory literature search and expert interviews (phase 1: March–September 2021) followed by a survey using the Delphi method (phase 2: March–July 2022). Experts interviewed in phase 1 of the study comprised stakeholders in leading positions in hospitals, hospital associations, research institutions in the field of quality and efficiency in health care, statutory health insurance, and patient representatives. In this article, the results of phase 2 are reported. The methodology and results of phase 1 will be published elsewhere. The study was conducted and is reported in accordance with the “Guidance on Conducting and Reporting Delphi Studies” (CREDES) [12].

### 2.1. The Delphi Study

The Delphi method is a structured group communication process that enables a group of experts in a specific field to engage with a complex problem [13]. Anonymous responses from participating experts are summarized and provided as feedback between multiple survey rounds, allowing the participants to revise their responses. Stopping criteria used to determine the end of the Delphi survey include reaching consensus, stability of responses, and/or a predefined number of rounds [12,14,15]. This study used a policy Delphi approach to examine a policy issue, thus providing relevant information for informed decision-making on policy options [16,17,18]. The objectives of a policy Delphi are to raise all relevant options and consequences of a policy (here: actions and practices related to the implementation of VBC), to determine the degree of consensus or disagreement as well as to assess the acceptability of the different options [19,20,21]. As such, in contrast to the traditional Delphi method, reaching a consensus is not a primary aim of a policy Delphi study [18,19]. The survey in this study was conducted for a predefined number of two rounds. Moreover, the approach taken in this study utilized a modified Delphi method in that the traditional qualitative first round with open-ended questions to identify issues and initial statements was replaced with a literature search and semi-structured expert interviews (phase 1). The results of phase 1 informed the development of the survey instrument as described below. Figure 1 displays the stages of the research process used in this study.

### 2.2. Development of the Survey

The development of the survey was based on the six core strategies formulated by Porter and Teisberg [3]. Additionally, a seventh strategy for improving treatment outcomes was identified from the qualitative data collected in phase 1 (i.e., semi-structured expert interviews). The seventh strategy addressed the *improvement of transparency* in the healthcare system. The need for improving transparency included transparency regarding responsibilities and decision-making bodies within the highly fragmented German healthcare system. Based on the key aspects raised in these interviews, the survey instrument was developed and implemented in the online survey tool LimeSurvey (Version 3.28.14 + 220608). In total, 56 actions and practices for implementing VBC were phrased which corresponded to the seven strategies described above. Accordingly, the survey was structured into seven blocks of questions, each on one strategy. Participants were asked to evaluate the relevance as well as the feasibility of each of the 56 provided actions and practices for implementing VBC in Germany, resulting in 112 survey items. The level of agreement with the perceived relevance and feasibility of the respective actions and practices was assessed using a four-point Likert scale (1 = ‘agree’, 2 = ‘somewhat agree’, 3 = ‘somewhat disagree’, 4 = ‘disagree’). After pilot testing the survey instrument among individuals from the target group (i.e., experts involved and/or affected by implementing VBC, n = 3) and obtaining feedback from other researchers in the field, adaptions were made. These included the alignment of wording, correction of spelling mistakes, and revisions regarding the order of questions. At the end of each block of questions on one strategy, free text fields were included for participants to give reasons for the answers they were unsure of, as well as to provide comments and ideas for further specification of respective strategies. This information was aimed at the purpose of deriving new questions for use in the subsequent survey round or modifying existing questions [13,18].

### 2.3. The Panel Members

Panelists were selected using purposive sampling. Panel members were considered suitable if they were either directly involved—on a strategic level—in the professional organizations of care provider groups, the remuneration of care, or if they were involved in the conceptual work regarding legislation, measurement, or evaluation of the quality of care. Potential panel members were recruited purposefully based on key stakeholders in the German healthcare system previously defined [22] and identified via the websites of the respective healthcare-related institutions. Particularly, senior positions of the respective institutions were identified, as they were assumed to have sufficient experience and overview of the healthcare system. Invitations were sent to potential participants (n = 416) via emails containing the link to the online survey, followed by two reminder emails at weekly intervals. A time frame of three weeks was set for the completion of the survey, after which the collection of responses ceased. Four weeks after the end of the first survey round, the same procedure was followed for the second survey round. No specific knowledge of the VBC concept was required as the general concept was outlined on an introductory page of the questionnaire and explanations of the individual strategies were provided at the beginning of each strategy-specific section. Moreover, links to additional information were included in the questionnaire as well as the invitation email. To start the survey, participants were required to declare informed consent. Data collection was fully anonymous. The complete survey instrument can be reviewed in the Appendix A. Ethical approval to conduct the trial was obtained from the University Hospital of Cologne Research Ethics Committee (ID: 20-1590_1).

### 2.4. Data Analysis and Definition of Consensus

The level of consensus was calculated based on the proportion of ratings in each level of agreement. Consensus levels were defined a priori according to de Loë [23] who differentiated a “high”, “moderate”, or “low” level of consensus. In addition, the group response was classified as “dissent” if the response pattern led to an ambiguous result. Cut-off values for consensus levels are displayed in the subscript of Table 2. Additionally, the direction of consensus was determined (if consensus occurred), which can be in favor (Likert scale levels “agree” and “somewhat agree”) or against (Likert scale levels “somewhat disagree” and “disagree”) the presented actions and practices for implementing VBC. 

In the subsequent survey, round only actions and practices for which responses of the initial survey round resulted in low consent or dissent were included for re-evaluation. Panel members were provided with the results of the initial survey round so they could reconsider their responses in the second round based on the overall group response. 

The two survey rounds were quantitatively analyzed using descriptive statistics using the public domain statistical software R 4.1.2 [24]. For this, measures of central tendency (median, mean, standard deviation (SD)) were calculated in addition to determining the level and direction of consensus as described above. Moreover, items were ranked in order of their mean relevance and feasibility. In addition to item-level results, strategy-level results were obtained by averaging item percentage values.

## 3. Results

From the 416 members of the target group invited to participate, 51 panel members provided complete survey data in the first round of the survey and an additional 18 individuals partially completed the questionnaire. In the second survey round, 43 panel members filled in the questionnaire completely and nine additional respondents provided incomplete data. Demographic characteristics of the panel members are displayed in Table 1. In both survey rounds the majority of respondents were female (>60%). Age distributions were comparable between survey rounds with a slightly higher share of respondents in the age range of 30–39 years in the second survey round. Panel members represented a wide range of educational and professional backgrounds, with the largest groups being representatives of hospitals and related associations, statutory health insurance and related associations, as well as members of representations of interests of other service providers.

In round 1 of the survey, consensus regarding relevance was found for 51 of the 56 (91%) actions and practices assessed. In terms of feasibility, 37 of the 56 (66%) actions and practices reached consensus. Table 2 displays the results for all questionnaire items after two survey rounds. For items that were re-rated in the second survey round, only results from the second round are shown (for separate results of the individual survey rounds, see Appendix A). Comments made in the free text fields contained general remarks (e.g., “Reform of remuneration systems urgently needed”, “These measures would imply a complete redesign of the health system”). No further survey questions could be derived from this information. In the second survey round, panel members re-evaluated those actions and practices, for which weak consensus or dissent was found in survey round 1. This applied to 43 actions and practices (nine items regarding relevance and 34 items regarding feasibility). 

After the second survey round, consensus was reached on six of the nine (67%) re-evaluated actions and practices in terms of relevance and on 28 of the 34 (82%) re-evaluated items in terms of feasibility. 

In total, after two survey rounds, consensus was reached for 53 (95%) of the 56 items in terms of relevance and for 50 (89%) of the 56 items regarding feasibility. The level of consensus after two survey rounds was high for 39 (70%) and 29 (52%) of the 56 items regarding relevance and feasibility respectively. A moderate level of consensus was found for 13 (23%) and 12 (21%) items and a low level of consensus was present in one (2%) and nine (16%) of the 56 items regarding relevance and feasibility, respectively.

Strategy-level aggregated agreement ratings for relevance as well as feasibility after two survey rounds are displayed in Figure 2. The highest share of agreement ratings in favor of the strategy in question was found for the relevance of strategies ‘*(6) Establishment of an information technology*’ and ‘*(7) Improve transparency*’. Likewise, these strategies ranked highly in terms of feasibility, yet agreement ratings on feasibility did not reach the same levels as for relevance. This is particularly evident in strategy 6. Strategy-level aggregated shares of agreement ratings of strategy *(1) Measurement of treatment outcomes and costs for every patient* ranked in the middle of all strategies. However, it should be noted that differences are evident within this strategy between actions and practices that relate to measuring outcomes and items corresponding to collecting data on costs. Higher agreement ratings were found for measuring outcomes compared to collecting data on costs both in terms of relevance and feasibility. These differences are not reflected in the strategy-level aggregated results as displayed in Figure 2 but are illustrated in Table 2. Overall, the share of ratings in favor of actions and practices was lowest both in terms of relevance and feasibility for strategy ‘*(5) Common remuneration of all treatment steps*’.

Item-level results after two survey rounds show that the responses of the expert panel were in favor of the presented actions and practices in 101 (98%) of the items for which consensus occurred. Opposition (moderate level of consensus) was found regarding the relevance of health care being provided preferably at one location for each indication (strategy ‘*(2) Organization of care in integrated care facilities & networks*’). Additionally, the panel opposes (moderate level of consensus) the feasibility of inter-sectoral joint budgets contingent on treatment outcomes achieved (strategy ‘*(5) Common remuneration of all treatment steps*’).

As displayed in Table 2, no consensus was found for either relevance or feasibility in a total of nine (8%) actions and practices. These nine items comprised three items covering the relevance domain while the remaining six items corresponded to the feasibility of the presented actions and practices. In detail, the panel was discordant both in terms of relevance and feasibility towards the suggestion of democratically electing the institution responsible for the determination, collection, and evaluation of treatment outcomes and costs (strategy ‘*(1) Measurement of treatment outcomes and costs for every patient*’). The panel was also inconclusive regarding the feasibility of health care being financed through joint cross-sector payments within a care network as well as care being provided for each indication preferably at one location (strategy ‘*(2) Organization of care in integrated care facilities & networks*’). Additionally, no consensus was reached on the feasibility of participating care institutions being responsible for expanding collaborations within care networks (strategy ‘*(4) Geographic expansion of excellent forms of care*’). Another area, in which the expert panel did not agree, covered reimbursement of care (strategy ‘*(5) Common remuneration of all treatment steps*’). In this respect, experts were ambiguous on the feasibility of risk adjusting the cross-sector joint budgets provided to a care network. Moreover, there was no consensus on the relevance of the cross-sector joint budget being based primarily on treatment outcomes achieved. As outlined above, for the feasibility domain of this item, there was opposition with a moderate level of consensus. Lastly, there was no consensus on the idea that treatment costs for preventable events (e.g., wound infection after surgery) should not be additionally reimbursed both in terms of relevance and feasibility.

In addition to levels and direction of consensus, mean agreement ratings for relevance and feasibility were assessed. Mean agreement ratings for the relevance of individual actions and practices ranged between 1.11 and 2.82, whereas mean agreement ratings for feasibility ranged from 1.57 to 2.96. The ten items with the highest mean agreement ratings on relevance and feasibility as well as their corresponding rank in the respective other domain are displayed in Table 3 and Table 4. As shown in Table 3, of the ten items that scored highest in terms of mean agreement on relevance, four corresponded to strategy ‘*(6) Building an enabling information technology platform*’. Specifically, the experts showed high levels of agreement in terms of relevance regarding the implementation of a digital patient record for each individual patient (M = 1.13). In connection with this, the panel highly agreed on the relevance of a standardized structure as a feature of patient records to enable collaboration across facilities (M = 1.11). Covering the entire course of treatment was another characteristic of the digital patient record that was highly agreed upon in terms of relevance (M = 1.17). The corresponding mean agreement ratings for the feasibility of these items are ranked in the second third of all the items evaluated, except for the item on digital patient records covering the entire course of treatment which was the third highest rated item in terms of feasibility.

As indicated by the strategy-level results reported above, the newly formulated strategy ‘(*7*) *Improving transparency*’ stands out in terms of agreement for both the relevance and feasibility domain.

On the level of individual items, the mean agreement ratings of four items in this strategy were among the ten items with the highest ranking in terms of relevance (Table 3). In detail, the expert panel considered improvements in transparency to be relevant at the micro-level in terms of patient-comprehensible communication (M = 1.20), structured shared decision-making (M = 1.27), as well as the presentation of different treatment options to patients (M = 1.29). On the macro-level, clearer communication of decision criteria emerged as the item receiving the 8th highest mean agreement score in terms of relevance (M = 1.25). Apart from the implementation of structured shared decision-making, all these items were likewise found to be among the highest-ranked items in terms of feasibility (Table 4). The highest mean agreement rating for feasibility was found for the item on providing routine care at less costly sites (strategy ‘*(3) Organization of integrated service provision between facilities*) (M = 1.57). The corresponding mean agreement score for relevance was slightly lower (M = 1.64) and ranked in the second half of all actions and practices. Similarly, the item receiving the second highest mean agreement score on feasibility (M = 1.58), namely collecting outcomes using valid instruments (strategy ‘*(1) Measurement of treatment outcomes and costs for every patient*), was ranked 23rd of all items in terms of mean agreement on relevance (M = 1.49).

## 4. Discussion

This Delphi study aimed to assess different stakeholders’ perceptions and to identify expert consensus on the relative relevance and perceived feasibility of actions and practices for moving towards a value-based healthcare system in Germany. Panel members were chosen via purposive sampling and covered a variety of professional fields and educational backgrounds, thus representing a range of different perspectives and diverse opinions towards the implementation of VBC. The German healthcare system is currently not prepared to implement VBC. In particular, the highly fragmented organization and delivery of health care, with strict separation of sectors, is seen as one of the most challenging structural barriers to the implementation of a value-based system in Germany [25,26]. Despite this, the results of this study show that stakeholders generally have a positive attitude towards the implementation of VBC in Germany. The majority of presented actions and practices for implementing VBC were agreed upon regarding their relevance for improving patient care. The expert panel also agreed on the feasibility of most of the actions and practices, yet the level of consensus and mean agreement ratings were lower compared to the relevance domain. 

The highest mean agreement ratings in terms of relevance were found for items corresponding to the implementation of a digital information platform. Levels of consensus were high for all five measures of this strategy regarding relevance. In total, four items related to this strategy were included in the top ten items with the highest mean agreement scores in terms of relevance. Likewise, consensus was found for the feasibility of all actions and practices for the implementation of a digital information platform, although mean agreement ratings did not reach the same high levels as for the relevance domain. Currently, the level of digitalization in the German health system is considerably lower compared to other countries [27]. Yet, the results of this study indicate that stakeholders generally share a positive position towards innovations in digital infrastructure which can be seen as a favorable foundation for further implementation of change in this domain. This positive attitude towards digitalization is particularly favorable as digital infrastructure is a key enabler for a number of other activities related to the implementation of VBC, such as organizing care in interprofessional teams within IPUs and measuring outcomes and costs [4,28]. 

A new strategy was derived from the expert interviews performed prior to the Delphi survey, covering the improvement of transparency in the healthcare system. There was unanimous agreement among the expert panel on the relevance of all the actions and practices of this newly formulated strategy. As there were also high levels of agreement on the feasibility of most actions and practices presented in this strategy, improving transparency in the healthcare system was identified as an area in which the expert panel perceived actions for change as approachable. The need to increase transparency perceived by the expert panel may have arisen from the particularly fragmented organization of the German healthcare system. 

The overall positive attitude towards actions and practices relevant to implementing VBC is supported by the finding that there were only two measures that were unanimously rejected by the expert panel. The direction of consensus was against the concept’s suggestion that care for each indication within an IPU should be provided at one location, if possible. This is intended to promote communication, teamwork, and as a result, efficiency for patients [4]. There was moderate consensus that stakeholders of the German healthcare system perceive this as not relevant. Additionally, there was dissent on the feasibility of this measure. Yet, co-locating the IPU team at one location is, according to the concepts’ authors, not seen as a critical component within the VBC concept, but rather as a supportive measure for the implementation of IPUs [4]. Overall, however, the expert panel agreed on the relevance of the general concept of IPUs. 

The second measure that was rejected by the panel concerns the feasibility of bundled reimbursement contingent on health outcomes achieved. In Germany, some form of bundled payment is already being implemented in inpatient care through Diagnosis Related Groups (DRGs). However, in contrast to the VBC concept, DRGs do not cover the full care cycle and do not take into account the health outcomes achieved [29]. Further discussion of this is provided below, as reimbursement likewise emerged as a recurring theme among the items for which no consensus was reached. Despite the finding that stakeholders do not consider bundled payments linked to health outcomes achieved as feasible, there was strong consensus on the relevance and feasibility of routinely measuring health outcomes data, which can be seen as an important prerequisite for value-based reimbursement. Measuring and reporting health outcomes achieved in connection with costs is described as the single most important action within the VBC concept to drive system change by creating competition among care providers for the best outcomes [4,25]. Although there was a high level of consensus for routinely measuring health outcomes for each patient, there was a moderate and low level of consensus for relevance and feasibility, respectively, for publishing this data. As others have noted, transparent reporting of provider performance in terms of health outcomes achieved by patients comes with the fear that “providers will discover outcomes they ignored, or outcomes they would prefer to ignore, or outcomes they would prefer others to ignore” [30]. Yet, transparent reporting of health outcomes data to both health care providers and the public, as proposed by the VBC concept, was confirmed to facilitate improvements in health outcomes achieved [31,32]. In this context, others have emphasized the need to ensure a fair process that includes, for example, methods for risk adjustment [33].

Porter and Guth point out that the health outcomes achieved do not need to be published in the first instance. Instead, they can initially be used as an internal database to identify areas for improvement [25]. In Germany, measurement of patient-relevant outcomes has been predominantly performed on a voluntary basis by provider initiatives and has been limited to specific indications, e.g., breast cancer, prostate cancer, or joint replacement [34]. Regulatory efforts aimed at routinely collecting patient-relevant outcomes data have only recently been initiated [35]. However, at this stage of implementation, the regulations for the mandatory collection of patient-relevant outcomes cover only a small selection of indications, are only performed retrospectively, the results are not publicly available, and data is collected only for a sample of 200 patients in each organization [34,35]. Yet, as international experience shows, government involvement in the process of introducing outcome measurement in routine care seems desirable to ensure standardization and alignment of measures across organizations [36]. As stakeholders in this study displayed readiness and acceptance towards routinely collecting patient-relevant outcomes for each patient, further expanding recent attempts seems advisable. Given that experts in this study were hesitant about the feasibility of publishing results, this should be attempted in a subsequent process, as recommended by Porter.

Measuring outcomes and costs are described as closely related actions within the VBC concept. In this study, actions and practices related to measuring and discussing resources expended received lower agreement ratings for both relevance and feasibility compared to measures related to collecting health outcomes. This stronger focus on outcomes with a tendency to neglect costs has been previously reported by others [37,38]. Different approaches for accurately measuring the costs of care for each patient have been developed [39]. In the context of VBC, time-driven activity-based costing (TDABC) has been described as the ‘gold standard’ to capture costs [40,41]. TDABC is becoming increasingly applied in different fields of healthcare with evidence of its facilitating effects on VBC recently emerging [39,41,42]. Further research and practical experience with the method will likely improve its applicability and thus acceptance by stakeholders of the healthcare system.

After two survey rounds, nine actions and practices remained on which the expert panel did not agree on either relevance (n = 3) or feasibility (n = 6). A total of five of these nine items correspond to aspects related to restructuring reimbursement models towards a value-based payment system. Value-based reimbursement aims at rewarding healthcare providers for maximizing value, e.g., enhanced health outcomes at lower costs [43]. The results of this study show that the expert panel did not reach a consensus regarding the feasibility of reimbursing health care by providing joint cross-sector payments to IPUs and care networks. Additionally, no unanimous group opinion was found on the feasibility of risk-adjusting joint bundled payments. Moreover, providing no additional reimbursement for preventable events was another measure whose feasibility was not agreed upon. Strikingly, the expert panel did also not agree on the relevance of one of the key aspects of value-based payment models, namely linking remuneration to health outcomes achieved. Furthermore, as described above, the expert panel unanimously rejected this measure in terms of feasibility. These results underline remuneration of care as the aspect in which the diverging interests of different stakeholders in the health system are most evident. Additionally, it became clear that stakeholders of the German system do not consider implementing reimbursement schemes as proposed by the VBC concept a realistic option today. Therefore, the results of this study suggest that other steps towards VBC should be taken first which will reinforce the subsequent implementation of reimbursement reforms. Taking into account the results of this study, the implementation of an enabling information technology system, actions to improve transparency within the healthcare system as well as the routine measurement of patient-relevant outcomes appear as promising initial measures. In this way, conditions can be modified in a positive way to facilitate the implementation of value-based reimbursement. 

According to Porter and Teisberg [3], change towards a value-based system must evolve as a bottom-up initiative from within healthcare providers and institutions. However, Mjåset et al. [7] highlighted government involvement as a crucial enabler for implementing VBC whilst emphasizing the importance of proactively involving the medical community. Changes in remuneration schemes constitute one aspect that is not likely to be implemented by bottom-up initiatives due to its complexity and diverging interests and should thus be tackled by political leaders in close collaboration with meso- and micro-level stakeholders [7].

### Limitations

This study has several limitations. First, the presented views of self-selected panel members might differ from those of the initially contacted group who did not respond to the invitation to the survey. Nevertheless, the diverse demographic and professional characteristics of the participants suggest the inclusion of a variety of perspectives in the results. As not all panel members provided answers to all the questions in the survey, sample sizes varied between questions. Since participants were given the option of free-text comments on the questions asked, non-responses to individual questions are probably due to the time or effort constraints associated with the lengthy survey rather than comprehension problems. Furthermore, some individuals did not participate in the second survey round leading to a reduced sample size in round two. However, as shown in Table 1, the round 2 panel covered a similar range of professional fields and educational backgrounds as was the case in round 1. Hence, it is unlikely that opinions and perspectives expressed in round 2 were affected by loss-to-follow-up. Moreover, as data collection was completely anonymous, it was not possible to ensure that the same individuals who filled in the first questionnaire also answered the second questionnaire as the Ethics Committee has prohibited storing the email addresses used to fill in the survey. This is especially relevant in those cases in which only general organization-wide e-mail addresses were available for invitations. In most cases, however, the invitation emails were sent directly to a representative within the institution. Lastly, the feasibility of actions and practices of the different core strategies of VBC are interconnected. It is therefore uncertain to what extent the panel members included assumptions about the implementation of other actions and practices in their assessment of the feasibility of one measure in question.

## 5. Conclusions

The results of this study show that German stakeholders are generally positive about implementing VBC to improve patient care in Germany. It is widely recognized that the implementation of VBC needs to be adapted to the conditions in the health system in question. Hence, as a next step in moving from the general notion of VBC as described by Porter towards a comprehensive implementation strategy in the German context, specific and tangible measures must be developed. The results of this study on stakeholder agreement on the relative importance and feasibility of VBC components should be considered in the already ongoing process of identifying and implementing initial activities to strengthen VBC in Germany. By doing so, changes in regulations are ensured to fit with stakeholders’ values, leading to improved acceptance and more successful implementation. The extent to which VBC is implemented in the German healthcare system needs to be determined in future studies.

## Figures and Tables

**Figure 1 healthcare-11-01187-f001:**
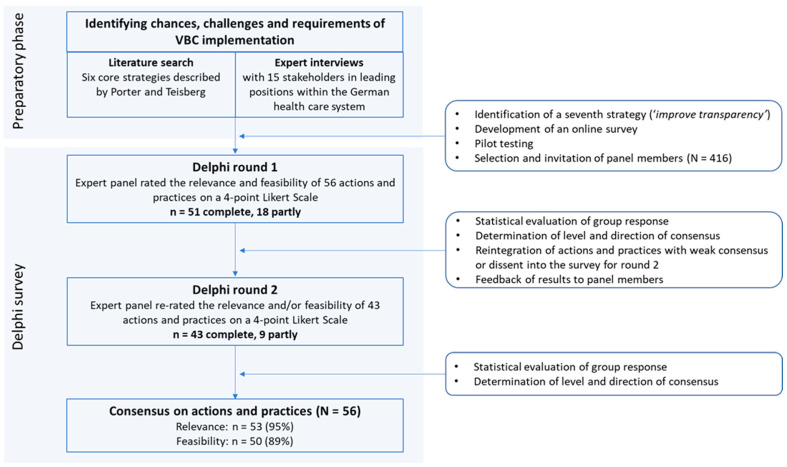
Stages of the Delphi process.

**Figure 2 healthcare-11-01187-f002:**
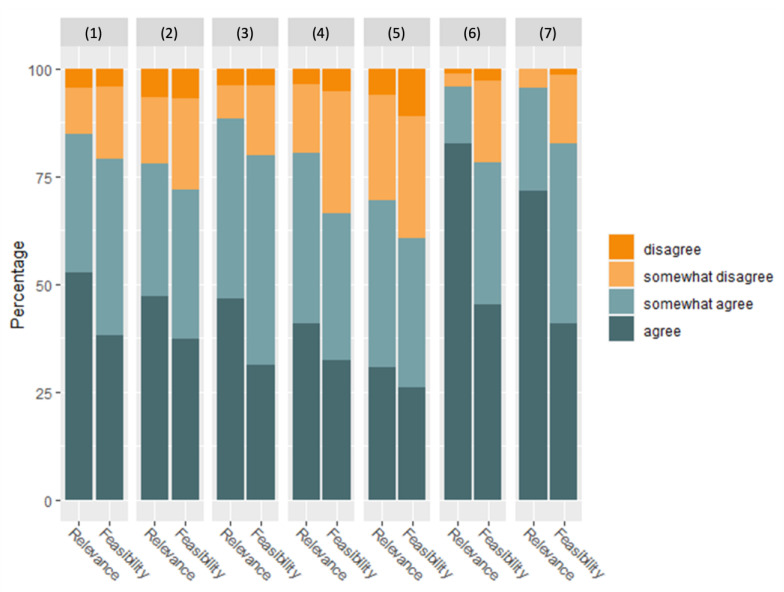
Percentages of agreement ratings for relevance and feasibility of VBC core strategies. Item percentage values were averaged to obtain strategy percentage values. Strategies: (1) Measurement of treatment outcomes and costs for every patient; (2) Organization of care in integrated care facilities & networks; (3) Organization of integrated service provision between facilities; (4) Geographic expansion of excellent forms of care; (5) Common remuneration of all treatment steps; (6) Establishment of an information technology; (7) Improve transparency.

**Table 1 healthcare-11-01187-t001:** Demographic characteristics of the panel members.

Delphi Survey	Round 1		Round 2	
	n = 51		n = 43	
	n	%	n	%
**Gender**				
Male	19	37%	16	37%
Female	31	61%	27	63%
**Age**				
20–29 years	2	4%	0	0%
30–39 years	9	18%	12	28%
40–49 years	11	21%	6	14%
50–59 years	21	41%	16	37%
60–69 years	8	16%	9	21%
	n	% *	n	% *
**Work area**				
Hospital and corresponding associations	15	29%	19	44%
Statutory health insurance and corresponding associations	10	20%	6	14%
Physician’s representation of interests	5	10%	1	2%
Medical management in a care facility	3	6%	4	9%
Commercial management in a care facility	2	4%	0	0%
Nursing management in a care facility	0	0%	2	5%
Pharmacies’ representation of interests	3	6%	2	5%
Patient representation	2	4%	1	2%
Institution of quality assessment in health care	2	4%	1	2%
Public administration (e.g., at federal, state, or local level)	1	2%	2	5%
Further area	21	41%	9	21%
Representation of interests of other service providers	19	37%	8	19%
Science	2	4%	0	0%
Advice center	0	0%	1	2%
**Educational background**				
Health Economics	10	20%	6	14%
Medicine	9	18%	9	21%
Health care and nursing	5	10%	4	9%
Pharmacy	5	10%	2	5%
Midwife	4	8%	3	7%
Social Sciences	4	8%	2	5%
Law	2	4%	2	5%
Psychology	2	4%	1	2%
Other care profession (e.g., logopedics, ergotherapy, physiotherapy)	16	31%	16	37%
Other educational background	17	33%	3	7%
Economics	5	10%	6	14%
Public Health	4	8%	1	2%
Health Sciences and Management	2	4%	1	2%
Other	4	8%	1	2%

* Percentage values do not add up to 100% as multiple selections were possible.

**Table 2 healthcare-11-01187-t002:** Results of the two rounds of the Delphi survey.

	Items *	Round	Distribution (in %)				Central Tendency			Consensus	
			Agree	Somewhat Agree	Somewhat Disagree	Disagree	Mean	SD	Median	Direction	Level
	**(1) Measurement of treatment outcomes and costs for every patient**										
Relevance	Data on outcomes tangible and important to patients	1	54	36	7	3	1.59	0.75	1	+	high
	Outcome data collection indication-specific	1	43	38	15	4	1.81	0.85	2	+	high
	Outcome data collection standardized	1	59	29	6	6	1.59	0.85	1	+	high
	Outcome data collection integrated into daily care	1	54	28	15	3	1.66	0.84	1	+	high
	Outcome data collection with valid instruments	1	57	38	3	1	1.49	0.63	1	+	high
	Outcome data on short- and long-term effects of treatment course	1	59	35	6	0	1.47	0.61	1	+	high
	All providers publish outcome data	1	40	31	19	10	2.00	1.01	2	+	moderate
	Patient-individual collection of financial resources expended	2	31	42	15	12	2.08	0.97	2	+	moderate
	Evaluation of expended financial resources by clinical and administrative staff	2	42	37	15	6	1.85	0.89	2	+	moderate
	Regular team meetings on outcome data	1	71	26	3	0	1.32	0.53	1	+	high
	Regular team meetings on expended resources	1	44	29	19	7	1.90	0.96	2	+	moderate
Feasibility	Data on outcomes tangible and important to patients	1	33	43	19	4	1.94	0.84	2	+	moderate
	Outcome data collection indication-specific	1	32	50	12	6	1.91	0.82	2	+	high
	Outcome data collection standardized	2	52	29	15	4	1.71	0.87	1	+	high
	Outcome data collection integrated into daily care	2	52	29	15	4	1.71	0.87	1	+	high
	Outcome data collection with valid instruments	2	52	40	6	2	1.58	0.70	1	+	high
	Outcome data on short- and long-term effects of treatment course	2	38	48	10	4	1.79	0.78	2	+	high
	All providers publish outcome data	2	37	31	27	6	2.02	0.94	2	+	low
	Patient-individual collection of financial resources expended	2	25	38	23	13	2.25	0.99	2	+	low
	Evaluation of expended financial resources by clinical and administrative staff	2	38	29	25	8	2.02	0.98	2	+	low
	Regular team meetings on outcome data	2	42	44	12	2	1.73	0.74	2	+	high
	Regular team meetings on expended resources	2	33	35	33	0	2.00	0.82	2	+	low
	Characteristics of institution responsible for determination, collection and evaluation of treatment outcomes and costs										
Relevance	Independent/neutral/free from conflicts of interest	1	84	13	3	0	1.19	0.47	1	+	high
	Legitimate members	1	40	35	18	7	1.93	0.94	2	+	moderate
	Legally defined tasks	1	43	41	16	0	1.74	0.73	2	+	high
	Equal representation (e.g., payers, service providers, patients)	1	54	26	9	10	1.75	1.00	1	+	high
	Democratically elected	2	33	25	27	15	2.25	1.08	2		dissent
	Interdisciplinary	1	69	26	4	0	1.35	0.57	1	+	high
	Scientific and methodological expertise	1	59	38	3	0	1.44	0.56	1	+	high
	Scientific-clinical expertise	1	66	31	1	1	1.38	0.60	1	+	high
Feasibility	Independent/neutral/free from conflicts of interest	2	52	29	19	0	1.67	0.79	1	+	high
	Legitimate members	1	32	49	15	4	1.91	0.81	2	+	high
	Legally defined tasks	1	32	54	13	0	1.81	0.65	2	+	high
	Equal representation (e.g., payers, service providers, patients)	1	29	46	19	6	2.01	0.86	2	+	moderate
	Democratically elected	2	25	35	27	13	2.29	1.00	2		dissent
	Interdisciplinary	1	38	49	9	4	1.79	0.78	2	+	high
	Scientific and methodological expertise	1	40	49	12	0	1.72	0.67	2	+	high
	Scientific-clinical expertise	1	43	50	7	0	1.65	0.62	2	+	high
	**(2) Organization of care in integrated care facilities & networks**										
Relevance	Multidisciplinary treatment (outpatient, inpatient & rehabilitative services)	1	80	18	2	0	1.22	0.45	1	+	high
	Indication-specific health care organization	1	50	30	13	7	1.77	0.93	1.5	+	high
	Health care as joint responsibility of multidisciplinary treatment team	1	68	20	8	3	1.47	0.79	1	+	high
	Health care planned from the outset	1	58	35	3	3	1.52	0.72	1	+	high
	Common management structure within a care network	1	37	33	22	8	2.02	0.97	2	+	moderate
	Common scheduling system within a care network	1	40	52	7	2	1.70	0.67	2	+	high
	Joint cross-sector payment within a care network	1	45	32	15	8	1.87	0.96	2	+	moderate
	Management by one team leader per patient within a care network	1	35	37	22	7	2.00	0.92	2	+	moderate
	Health care for each indication at one location	1	10	20	48	22	2.82	0.89	3	-	moderate
Feasibility	Multidisciplinary treatment (outpatient, inpatient & rehabilitative services)	2	50	31	17	2	1.71	0.82	1.5	+	high
	Indication-specific health care organization	2	44	42	13	2	1.73	0.76	2	+	high
	Health care as joint responsibility of multidisciplinary treatment team	2	44	31	21	4	1.85	0.90	2	+	moderate
	Health care planned from the outset	2	44	44	10	2	1.71	0.74	2	+	high
	Common management structure within a care network	2	29	31	27	13	2.23	1.02	2	+	low
	Common scheduling system within a care network	2	40	42	10	8	1.88	0.91	2	+	high
	Joint cross-sector payment within a care network	2	38	19	31	13	2.19	1.08	2		dissent
	Management by one team leader per patient within a care network	2	33	42	19	6	1.98	0.89	2	+	moderate
	Health care for each indication at one location	2	15	29	44	13	2.54	0.90	3		dissent
	**(3) Organization of integrated service provision between facilities**										
Relevance	Specific service offering of each care institution	1	46	39	8	7	1.76	0.88	2	+	high
	Disease-specific interdisciplinary providers with high treatment volume for scheduled or complex treatments	1	61	36	3	0	1.42	0.56	1	+	high
	Routine care provision at less costly sites	1	54	32	8	5	1.64	0.85	1	+	high
	Coordinating institution for cooperation between care institutions	2	26	60	11	4	1.94	0.73	2	+	high
Feasibility	Specific service offering of each care institution	2	23	57	19	0	1.96	0.66	2	+	high
	Disease-specific interdisciplinary providers with high treatment volume for scheduled or complex treatments	1	29	47	19	5	2.00	0.83	2	+	moderate
	Routine care provision at less costly sites	2	49	45	6	0	1.57	0.62	2	+	high
	Coordinating institution for cooperation between care institutions	2	23	45	21	11	2.19	0.92	2	+	low
	**(4) Geographic expansion of excellent forms of care**										
Relevance	Expanding excellent forms of care rather than the catchment area	1	29	55	13	4	1.91	0.75	2	+	high
	Cooperations in the form of care networks	1	66	30	4	0	1.38	0.56	1	+	high
	Care institutions responsible for expansion of collaboration	2	30	34	30	6	2.13	0.92	2	+	low
	Rotation of individual employees between participating care facilities	1	39	38	18	5	1.89	0.89	2	+	moderate
Feasibility	Expanding excellent forms of care rather than the catchment area	2	21	45	30	4	2.17	0.82	2	+	low
	Cooperations in the form of care networks	2	49	36	11	4	1.70	0.83	2	+	high
	Care institutions responsible for expansion of collaboration	2	26	28	38	9	2.30	0.95	2		dissent
	Rotation of individual employees between participating care facilities	2	34	28	34	4	2.09	0.93	2	+	low
	**(5) Common remuneration of all treatment steps**										
Relevance	Inter-sectoral, risk-adjusted joint budget provided to a care network	2	20	53	24	2	2.09	0.73	2	+	moderate
	Inter-sectoral joint budget for indications with multidisciplinary treatment needs	2	31	44	22	2	1.96	0.80	2	+	moderate
	Annually adjusted inter-sectoral joint budget	1	39	37	11	13	1.98	1.02	2	+	moderate
	Inter-sectoral joint budget based on treatment outcomes	2	27	29	42	2	2.20	0.87	2		dissent
	No reimbursement of costs for preventable events	2	13	40	36	11	2.44	0.87	2		dissent
	Additional reimbursement of costs for unavoidable events	1	54	30	11	6	1.69	0.89	1	+	high
Feasibility	Inter-sectoral, risk-adjusted joint budget provided to a care network	2	18	33	38	11	2.42	0.92	2		dissent
	Inter-sectoral joint budget for indications with multidisciplinary treatment needs	2	31	51	16	2	1.89	0.75	2	+	high
	Annually adjusted inter-sectoral joint budget	2	36	31	29	4	2.02	0.92	2	+	low
	Inter-sectoral joint budget based on treatment outcomes	1	7	20	41	31	2.96	0.91	3	-	moderate
	No reimbursement of costs for preventable events	2	18	33	33	16	2.47	0.97	2		dissent
	Additional reimbursement of costs for unavoidable events	2	47	38	13	2	1.71	0.79	2	+	high
	**(6) Establishment of an information technology**										
Relevance	Digital patient record for each patient	1	89	9	2	0	1.13	0.39	1	+	high
	Data relevant to care covers the entire course of treatment	1	87	11	0	2	1.17	0.51	1	+	high
	Standardized structure of digital patient record	1	91	8	2	0	1.11	0.38	1	+	high
	Digital patient record accessible to all providers involved in care	1	83	13	4	0	1.21	0.49	1	+	high
	Digital patient record as intelligent system with disease-specific recommendations	1	64	25	8	4	1.51	0.80	1	+	high
Feasibility	Digital patient record for each patient	1	40	34	26	0	1.87	0.81	2	+	moderate
	Data relevant to care covers the entire course of treatment	2	59	25	14	2	1.59	0.82	1	+	high
	Standardized structure of digital patient record	1	38	36	21	6	1.94	0.91	2	+	moderate
	Digital patient record accessible to all providers involved in care	1	38	36	25	2	1.91	0.84	2	+	moderate
	Digital patient record as intelligent system with disease-specific recommendations	2	52	34	9	5	1.66	0.83	1	+	high
	**(7) Improve transparency**										
Relevance	Clear communication of responsibilities in macro-level decisions	1	61	29	10	0	1.49	0.67	1	+	high
	Clear communication of decision criteria in macro-level decisions	1	75	25	0	0	1.25	0.44	1	+	high
	Clear communication of data bases in macro-level decisions	1	71	24	6	0	1.35	0.59	1	+	high
	Clear communication of decision-making bodies in macro-level decisions	1	65	27	8	0	1.43	0.64	1	+	high
	Clear communication of conflicts of interest in macro-level decisions	1	69	27	4	0	1.35	0.56	1	+	high
	Patient-comprehensible communication in micro-level decisions	1	82	16	2	0	1.20	0.45	1	+	high
	Presentation of different treatment options to patients in micro-level decisions	1	73	25	2	0	1.29	0.50	1	+	high
	Structured shared decision making in micro-level decisions	1	76	20	4	0	1.27	0.53	1	+	high
	Structured advice on self-management and health promotion in micro-level decisions	1	75	20	6	0	1.31	0.58	1	+	high
Feasibility	Clear communication of responsibilities in macro-level decisions	1	35	41	22	2	1.90	0.81	2	+	moderate
	Clear communication of decision criteria in macro-level decisions	1	37	39	22	2	1.88	0.82	2	+	moderate
	Clear communication of data bases in macro-level decisions	1	35	51	12	2	1.80	0.72	2	+	high
	Clear communication of decision-making bodies in macro-level decisions	1	45	41	12	2	1.71	0.76	2	+	high
	Clear communication of conflicts of interest in macro-level decisions	1	43	27	24	6	1.92	0.96	2	+	moderate
	Patient-comprehensible communication in micro-level decisions	1	47	43	10	0	1.63	0.66	2	+	high
	Presentation of different treatment options to patients in micro-level decisions	1	43	43	14	0	1.71	0.70	2	+	high
	Structured shared decision making in micro-level decisions	1	41	43	16	0	1.75	0.72	2	+	high
	Structured advice on self-management and health promotion in micro-level decisions	1	41	45	14	0	1.73	0.70	2	+	high

* Displayed item descriptions represent short versions, for full-length items see Appendix A. Level of consensus [23]: ≥70% of ratings in one category or ≥80% in two contiguous categories = high consensus. ≥60% of ratings in one category or ≥70% in two contiguous categories = moderate consensus. ≥50% of ratings in one category or ≥60% in two contiguous categories = low consensus. <50% of ratings in one category or <60% in two contiguous categories = dissent.

**Table 3 healthcare-11-01187-t003:** The 10 items with highest mean agreement scores in terms of relevance.

Relevance Rank *	Strategy	Item	Mean Relevance	Mean Feasibility	Feasibility Rank *
1	(6) Establishment of an information technology	Standardized structure of digital patient record	1.11	1.94	36
2	(6) Establishment of an information technology	Digital patient record for each patient	1.13	1.87	26
3	(6) Establishment of an information technology	Data relevant to care covers the entire course of treatment	1.17	1.59	3
4	(1) Measurement of treatment outcomes and costs for every patient	Characteristics of institution responsible for determination, collection and evaluation of treatment outcomes and costs: independent/neutral/free from conflicts of interest	1.19	1.67	7
5	(7) Improve transparency	Patient-comprehensible communication in micro-level decisions	1.20	1.63	4
6	(6) Establishment of an information technology	Digital patient record accessible to all providers involved in care	1.21	1.91	31
7	(2) Organization of care in integrated care facilities & networks	Multidisciplinary treatment (outpatient, inpatient & rehabilitative services)	1.22	1.71	11
8	(7) Improve transparency	Clear communication of decision criteria in macro-level decisions	1.25	1.88	28
9	(7) Improve transparency	Structured shared decision making in micro-level decisions	1.27	1.75	20
10	(7) Improve transparency	Presentation of different treatment options to patients in micro-level decisions	1.29	1.71	10

* Rankings correspond to the 56 actions and practices presented.

**Table 4 healthcare-11-01187-t004:** The 10 items with highest mean agreement scores in terms of feasibility.

Feasibility Rank *	Strategy	Item	Mean Feasibility	Mean Relevance	Relevance Rank *
1	(3) Organization of integrated service provision between facilities	Routine care provision at less costly sites	1.57	1.64	29
2	(1) Measurement of treatment outcomes and costs for every patient	Outcome data collection with valid instruments	1.58	1.49	23
3	(6) Establishment of an information technology	Data relevant to care covers the entire course of treatment	1.59	1.17	3
4	(7) Improve transparency	Patient-comprehensible communication in micro-level decisions	1.63	1.20	5
5	(1) Measurement of treatment outcomes and costs for every patient	Characteristics of institution responsible for determination, collection and evaluation of treatment outcomes and costs: Scientific-clinical expertise	1.65	1.38	17
6	(6) Establishment of an information technology	Digital patient record as intelligent system with disease-specific recommendations	1.66	1.51	25
7	(1) Measurement of treatment outcomes and costs for every patient	Characteristics of institution responsible for determination, collection and evaluation of treatment outcomes and costs: independent/neutral/free from conflicts of interest	1.67	1.19	4
8	(4) Geographic expansion of excellent forms of care	Cooperations in the form of care networks	1.70	1.38	16
9	(7) Improve transparency	Clear communication of decision-making bodies in macro-level decisions	1.71	1.43	19
10	(7) Improve transparency	Presentation of different treatment options to patients in micro-level decisions	1.71	1.29	10

* Rankings correspond to the 56 actions and practices presented.

## Data Availability

The datasets used and analyzed in this study are available from the corresponding author on reasonable request.

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
