# Peer review of "Transforming Health Care Delivery towards Value-Based Health Care in Germany: A Delphi Survey among Stakeholders"

_healthcare, 2023, doi:10.3390/healthcare11081187_

Round 1
Reviewer 1 Report
The article focuses on a Delphi survey to explore stakeholders' views on the relevance and feasibility of actions and practices related to introducing value-based healthcare in the German healthcare system. The study's relevance is justified because value-based healthcare is one of the strategies for solving growing problems in healthcare systems. At the same time, value-based healthcare is not widely used in the German healthcare system today. Therefore, the authors conducted a Delphi survey to explore stakeholders' views on the relevance and feasibility of actions and practices related to introducing value-based healthcare in the German healthcare system. The panelists were selected using a purposive sample. Two iterative rounds of online surveys were conducted, preceded by a literature search and semi-structured interviews. After two rounds of the survey, the consensus was reached on 95% of the questions on relevance and 89% on feasibility. The panel responses favored the value-based healthcare practices presented in 98% of the questions on which consensus was reached (n = 101). Objections have been raised regarding the advisability of providing medical care preferably in one place for each indication. In addition, the panel considered intersectoral joint budgeting based on treatment outcomes inappropriate. When planning the next steps in the transition to a values-based health system, policymakers should consider the results of this study on stakeholder perceptions of the relative importance and feasibility of value-based health components. This ensures that regulatory changes align with stakeholder values, promoting wider acceptance and more successful implementation.
Despite the satisfactory quality of the article, some shortcomings need to be corrected.
- Including the section with an analysis of the current state of research is recommended.
- The participant's portrait in the study should be described in more detail.
- Describing the results presented in table 2 in more detail in the text is recommended.
- It should be briefly discussed why not all the participants answered all proposed questions.
- The practical use of the obtained results should be highlighted.
In summarizing my comments, I recommend that the manuscript is accepted after minor revision.
Author Response
Dear Reviewers,
Thank you for your thorough review, the positive feedback, and your helpful remarks to improve the article. Please find our point-by-point response below.
Reviewer 1
- Including the section with an analysis of the current state of research is recommended.
Response
Thank you for this suggestion. We included the following paragraph in the end of the introduction: “Even though implementation of VBC is already being advanced in other countries [5,8,9] or at the supranational level [10], little knowledge exists on implementation specifically in the German context. Previous literature examined the German healthcare providers’ knowledge on VBC, but only focused on the hospital sector [11]. There exists a knowledge gap regarding the perspective of various German healthcare stakeholders on specific steps of implementing the VBC approach.“
- The participant's portrait in the study should be described in more detail.
Response
The conceptual characteristics making an individual eligible to be invited for this survey are described in section 2.3. More specifically these individuals were individuals with leading positions in various institutions and organizations of the German health care system. The institutions and organizations included e.g. the ministry of health, association of Statutory Health Insurance Physicians, Statutory Health Insurances, public health authorities, Federal Joint Committee, Institute for Quality and Efficiency in Health Care, hospital associations. For a complete overview of the institutions and organizations that were invited, please see reference 22.
The characteristics of the individual people, who participated, are displayed in table 1.
- Describing the results presented in table 2 in more detail in the text is recommended.
Response
The paragraphs from line 211 to line 273 describe the results displayed in table 2. We are happy to extend the description based on your suggestion. For this, please let us know, which specific part or topic of the table should be elaborated in more detail.
- It should be briefly discussed why not all the participants answered all proposed questions.
Response
Thank you for directing us to this question. We adapted the section on limitations accordingly: “As not all panel members provided answers for all the questions of the survey, sample sizes varied between questions. Since participants were given the option of free-text comments on the questions asked, non-response to individual questions is probably due to time or effort constraints associated with the lengthy survey rather than comprehension problems.“
- The practical use of the obtained results should be highlighted.
Response
Please see “Conclusions” for thoughts on how to use the results. We think it is highly important to consider the findings when developing or refining activities to implement VBC in the German healthcare system. Acceptance from the wider community and those who are asked to implement specific measures, is crucial for successful and effective implementation of VBC:
For more details please see the revised version manuscript.
Reviewer 2 Report
This was a very interesting and relevant manuscript that created much thought on my behalf.
When considering the levels of consensus; high, moderate, low and dissent, how did you decide on the values for these, were they based on previous literature?
There was a large dependence on the work of Porter and others, are there any other authors that you could have referenced?
This was a very novel piece of work, that has relevance throughout the world. I imagine that other countries undertaking a similar study would find similar results/outcomes/conclusions.
Author Response
Dear Reviewers,
Thank you for your thorough review, the positive feedback, and your helpful remarks to improve the article. Please find our point-by-point response below.
Reviewer 2
- When considering the levels of consensus; high, moderate, low and dissent, how did you decide on the values for these, were they based on previous literature?
Response
Please see section 2.4 data analysis, first paragraph:
“Consensus levels were defined a priori according to de Loë [23] who differentiated a “high”, “moderate” or “low” level of consensus. In addition, the group response was classified as “dissent” if the response pattern led to an ambiguous result.“
- There was a large dependence on the work of Porter and others, are there any other authors that you could have referenced?
Response
Porter is the major author, who introduced the concept of VBC in healthcare. Together with various colleagues (e.g. Teisberg and Lee) he elaborated on the concept. There are analyses of the practical implementation of VBC, which are also referenced in the introduction. Apart from Porter and his collaborators we are not aware of another author or research group, who work on the conceptualization of VBC. However, we used a qualitative approach during the preparatory phase (see e.g. figure 1) to check whether all elements of VBC as suggested by Porter and colleagues, are relevant for the German context and whether additional elements should be considered. The detailed results of this preparatory phase are not yet published, but the key finding of an additional relevant element / strategy (Improving transparency) is already considered in the Delphi study.
For more details please see the revised version manuscript.
Reviewer 3 Report
The paper on the Delphi survey about introducing value-based health care in Germany is an interesting and timely paper on a health policy issue that is likely to become important in the near future. The study is well designed and well written and the bibliography is appropriate and informative. The authors followed a strict protocol in obtaining the views of the stakeholders that they surveyed, regarding the relevance and feasibility of the items in the questionnaire. They present the results in a clear and systematic way, and the discussion highlights the points that are important. The paper shows that consensus can be reached on most of the subjects raised by the study, regarding the introduction of value-based health care in Germany.
There is just one point that requires clarification: from the initial sample of 416 potential participants, who were selected using purposive sampling, 51 questionnaires were fully answered in the first round and 43 in the second round. Thus, the panel members whose views are presented were a self-selected group of persons among the initial 416 potential participants. It is possible that the views of this self-selected group differ from those who did not respond to the invitation to participate in the survey. This point should be added to the limitations since it affects how representative of the whole are those who did participate in the survey.
Author Response
Dear Reviewers,
Thank you for your thorough review, the positive feedback, and your helpful remarks to improve the article. Please find our point-by-point response below.
Reviewer 3
- There is just one point that requires clarification: from the initial sample of 416 potential participants, who were selected using purposive sampling, 51 questionnaires were fully answered in the first round and 43 in the second round. Thus, the panel members whose views are presented were a self-selected group of persons among the initial 416 potential participants. It is possible that the views of this self-selected group differ from those who did not respond to the invitation to participate in the survey. This point should be added to the limitations since it affects how representative of the whole are those who did participate in the survey.
Response
Thank you for raising this important limitation. We adapted the section on limitations based on your suggestion:
“First, the presented views of self-selected panel members might differ from those of the initially contacted group who did not respond to the invitation to the survey. Nevertheless, the diverse demographic and professional characteristics of the participants suggest the inclusion of a variety of perspectives in the results.“
For more details please see the revised version manuscript.